# Secreted miR-153 Controls Proliferation and Invasion of Higher Gleason Score Prostate Cancer

**DOI:** 10.3390/ijms23116339

**Published:** 2022-06-06

**Authors:** Gloria Bertoli, Antonella Panio, Claudia Cava, Francesca Gallivanone, Martina Alini, Giulia Strano, Federico Molfino, Loredana Brioschi, Paola Viani, Danilo Porro

**Affiliations:** 1Institute of Bioimaging and Molecular Physiology (IBFM)-CNR, 20090 Segrate, Italy; claudia.cava@ibfm.cnr.it (C.C.); francesca.gallivanone@ibfm.cnr.it (F.G.); martina.alini01@universitadipavia.it (M.A.); strano.giulia92@gmail.com (G.S.); molfinof@gmail.com (F.M.); danilo.porro@ibfm.cnr.it (D.P.); 2Department of Medical Biotechnology and Translational Medicine, Università degli Studi di Milano, 20054 Milan, Italy; loredana.brioschi@unimi.it (L.B.); paola.viani@unimi.it (P.V.)

**Keywords:** microRNA, prostate cancer (PC), diagnosis, miRNA secretion, exosomes, microvesicles

## Abstract

Prostate cancer (PC) is a male common neoplasm and is the second leading cause of cancer death in American men. PC is traditionally diagnosed by the evaluation of prostate secreted antigen (PSA) in the blood. Due to the high levels of false positives, digital rectal examination and transrectal ultrasound guided biopsy are necessary in uncertain cases with elevated PSA levels. Nevertheless, the high mortality rate suggests that new PC biomarkers are urgently needed to help clinical diagnosis. In a previous study, we have identified a network of genes, altered in high Gleason Score (GS) PC (GS ≥ 7), being regulated by miR-153. Until now, no publication has explained the mechanism of action of miR-153 in PC. By in vitro studies, we found that the overexpression of miR-153 in high GS cell lines is required to control cell proliferation, migration and invasion rates, targeting Kruppel-like factor 5 (*KLF5*). Moreover, miR-153 could be secreted by exosomes and microvesicles in the microenvironment and, once entered into the surrounding tissue, could influence cellular growth. Being upregulated in high GS human PC, miR-153 could be proposed as a circulating biomarker for PC diagnosis.

## 1. Introduction

PC is one of the most common neoplasms among men and is the third cause of cancer-related men death worldwide [1]. The screening of PC is mainly based on blood-based prostate secreted antigen (PSA) test examination. This test, introduced in the late 1980s, increased the chance of identifying PC at early stage, reducing PC-associated mortality, and thus suggesting the use of PSA level the in blood as a biomarker for PC diagnosis. Nevertheless, the percentage of false positive is high, and a large portion of benign conditions such as benign prostatic hyperplasia and prostatitis can be differentially diagnosed with respect to malignant neoplasm, using PSA levels. Furthermore, malignant PC is a heterogeneous disease characterized by a wide spectrum of clinical presentations and possible outcomes, and in a clinical context, differential diagnosis of PC characteristics is the main challenge to identify optimal and tailored treatment for each patient while limiting overtreatment and reducing mortality. For these reasons, in the last few years, there has been growing attention in finding reliable tools for a non-invasive assessment of PCa aggressiveness, in particular, identifying low-grade disease, which can be addressed by active surveillance without significant changes in the outcome.

At present, a clinical workup of patients suspected for PC following PSA dosage include diagnostic imaging, such as magnetic resonance imaging (MRI) [2], and PC characteristics are commonly established using the Gleason Score (GS) obtained by image-guided biopsy, which is an index that includes histologic aggressiveness, cellular differentiation status and glandular structure. Despite this, in more than 50% of cases, GS obtained by prostate biopsy specimens does not represent the true GS of the tumor, due to both biological cancer heterogeneity and sampling errors [3]. Recent works were dedicated to explore advanced image processing methodologies, such as radiomics, to predict tumor aggressiveness from advanced quantification of MRI images (e.g., [4]). Even if the results suggested that reliable radiomic signatures can be evaluated for prediction of PC aggressiveness in terms of Gleason score, extracapsular extension, and nodal stage, the published radiomic approaches have to be standardized and validated in large patient cohorts. Furthermore, even if minimally invasive, both MRI exam and image-guided biopsy are uncomfortable for the patient, in particular when repeated over time in cases of active surveillance, and are expensive procedures. For these reasons, the European Association of Urology (EAU), in 2021, considered new tools for the early diagnosis of PC, which were able to distinguish patients who do not need magnetic resonance imaging (MRI) or biopsy from those with more aggressive disease that requires further tests or treatments [5]. Thus, in a clinical context, the possibility to identify new biomarkers with rapid and inexpensive methods to support early diagnosis of significant or aggressive PCs, meaning those with GS value ≥7, and evaluation of PC development over time in case of active surveillance is still an open challenge.

MicroRNAs are a class of small, non-coding RNA molecules that are able to modulate the expression of specific target mRNA by pairing with a specific seed sequence. Several papers identified miRNAs as key regulators of a wide variety of oncogenic activities, such as proliferation, invasion, migration, apoptosis, stemness and differentiation, and metastasis (e.g., [6,7]). Several miRNAs are deregulated in PC compared to normal prostatic tissue [8,9], and alterations in the miRNA profile impact gene expression, thus promoting aggressive behavior of the cells and promoting PC development and metastasis. In recent years, several miRNAs have been described as potential predictive molecules, for recurrence (e.g., miR-21) [10], for metastasis presence (e.g., miR-182-5p) [11] and for poor prognosis (e.g., miR-17-5p) [12]. With an integrative approach, we identified a gene network differentially expressed in high GS PC tissues compared to normal tissues [9]. This network, composed of 19 genes, is highly regulated by *Hsa-miR-153* (miR-153). Deregulation of miR-153 has recently been observed in several human cancers, while miR-153 serves an oncogene or tumor-suppressive role in different cancer types [13]. Although the expression of miR-153 has already been associated with poor prognosis in PC [13], no publication explains the functional role of miR-153 in PC with high GS (GS ≥ 7). In this study, we analyzed the effect of miR-153 modulation in high GS PC cell lines, suggesting a possible target gene. Moreover, as miRNAs can be secreted in biofluids (blood, serum, plasma, urine, etc.) we investigated the possibility to quantify miR-153 in a culture medium and its effect on surrounding cells.

## 2. Results

### 2.1. miR-153 Is Overexpressed in PC Cells with Higher Gleason Score

RT-qPCR analysis revealed that miR-153 is significantly upregulated in PC cells (Figure 1A) with higher expression in GS 8 PC3 cells (*t* test, *p* value = 0.01) than in GS 7 DU145 (*t* test, *p* value = 0.005) cells compared to benign BPH1 cells.

UALCAN software [14] confirmed the overexpression of miR-153 in TCGA samples from prostate carcinoma (PRAD) compared to normal tissues (Figure 2A) and in different GS primary tissues starting from GS 6 up to GS 10 (Figure 2B). Moreover, the UALCAN analysis revealed that the higher expression of miR-153, although not significant, is associated with poor prognosis compared to its low expression (*p* = 0.66, figure not shown). We could explain the non-significant association between high miR-153 expression and poor prognosis because UALCAN software is based on the third quartile of miR-153 for the high-expression group. Thus, the limitation of UALCAN software could be the inability of the user to choose an optimal cut-off specific for their own data.

### 2.2. KLF5 Is a Direct Target of miR-153

Kruppel-like factor 5 (*KLF5*) is a predicted target of miR-153 by TargetScan and our previous analysis [15]. TargetScan revealed two poorly conserved sites (not shown) and one conserved site in the 3′ untranslated region (UTR) of *KLF5* mRNA sequence (Figure 3A). The RT-qPCR analysis of the expression of *KLF5* in PC cell lines revealed that DU145 and PC3 cell lines have a lower expression of *KLF5* in respect to benign BPH1 cells (Figure 4B; *t* test, *p* value = 0.012 and 0.047, respectively). Moreover, the silencing of miR-153, obtained by antisense (As) oligonucleotide miR-153 treatment (Figure 3C,E; *p* value = 0.03 for DU145, and *p* = 0.03 for PC3), results in upregulation of *KLF5* in both cell lines (Figure 3D, *p* value = 0.008 for DU145; and Figure 3F, *p* = 0.023 for PC3). In order to demonstrate that *KLF5* is a direct target of miR153, we cloned the seed sequence of human *KLF5* into a luciferase reporter construct. The results of the luciferase assay, performed in the presence of As miR-153, revealed an increase in relative light units (RLU) emission in both DU145 and PC3 cells compared to scr-treated cells, as shown in Figure 3G,H, respectively. Furthermore, the transfection of S miR-153 caused a decreased in RLU emission compared to scr-treated cells in both cell lines (Figure 3G,H, last column).

### 2.3. miR-153 Promotes Cell Proliferation in Higher GS Cells

In order to understand if miR-153 has a role in the control of PC cell proliferation, we modulated the expression of miR-153 by using 200 nM As miR-153 oligonucleotide in DU145 and PC3 cells (Figure 4A,B, respectively). In vitro reduction of miR-153 expression inhibits the proliferation of both DU145 and PC3 cells (Figure 4C,D, respectively) compared to scrambled-treated cells. This effect could also be visualized by checking the confluency of the cells into the seeding wells (Figure 5). The silencing of the cells did not affect the apoptotic process, as we did not observe a significant increase in the expression of pro-apoptotic mRNAs (i.e., Bad, Bax) or a decrease in anti-apoptotic genes (i.e., Bcl2) (Figure 5B–D).

### 2.4. miR-153 Regulates Cell Migration and Invasion of PC Cells

The activity of miR-153 on migration was analyzed by wound healing assay. The presence of As miR-153 reduced the migratory capability of both DU145 and PC3 (see Figure 6A,B, respectively), in comparison with both untreated and scrambled-treated cells. In addition, the invasion ability of the cells, treated by As miR-153, was affected, as suggested by Boyden’s chamber results in both DU145 and PC3 cells (Figure 7A,B, respectively).

### 2.5. miR-153 Can Be Secreted, Influencing Surrounding Tissue

Since several other miRNAs have been demonstrated to be secreted by cancer cells, we evaluated the hypothesis that miR-153 could be also released in the surrounding environment. To analyze the expression level of secreted miR-153, we collected the culture medium of PC3 and DU145 (conditional medium, CM) cell lines, isolated the released total RNAs and performed miR-153 analysis in RT-qPCR. The assay revealed that miR-153 is secreted by both DU145 and PC3 (Figure 8A), and that the relative expression of miR-153 in conditional medium is lower than the intracellular amount of miR-153 in both cell lines.

To understand if secreted miR-153 could enter surrounding cells, we placed the conditional medium of DU145 and PC3 cells onto the BPH1 cells. We cultured BPH1 cells, which have a lower expression of miR-153 compared to DU145 and PC3, in the CM isolated from DU145 and PC3 cells. After 48 *h* of culture in DU145 or PC3 CM, BPH1 showed increased expression of miR-153 compared to cells grown in their standard medium (Figure 8A; *t* test, *p* value = 0.014 for DU145 CM and 0.015 for PC3 CM), with higher expression of miR-153 in conditional medium from PC3 and DU145 cells compared to that of BPH1 cells (Figure 8B *t* test, *p* value = 0.017 for PC3 CM and 0.003 for DU145 CM). Culturing BPH1 cells in CM from DU145 or PC3 cells increased intracellular miR-153 expression levels, with a higher expression in BPH1 cultured in CM from DU145 (*t* test, *p* value = 0.014) compared to that from PC3 (*t* test, *p* value = 0.015) (Figure 8C).

The RT-qPCR analysis on mRNA content of DU145 CM- or PC3 CM-treated BPH1 revealed that, in both these samples, *KLF5* mRNA was reduced compared to untreated BPH1 cells (Figure 8D *t* test, *p* value = 0.032 for DU145 CM and 0.021 for PC3 CM).

In order to understand if the secreted miR-153, once entered into the BPH1 cells, could affect their behavior, we analyzed the proliferation of the BPH1 cells grown on a conditional medium from DU145 or PC3 cells. As shown in Figure 8E, the cells in conditional medium revealed a decrease in proliferation rate (*p* value = 0.032 for BPH1+CM of PC3 compared to untreated BPH1; *p* value = 0.016 for BPH1+CM of DU145 compared to untreated BPH1).

### 2.6. Exosomal and Microvesicle Distribution of Secreted miR-153

In order to understand if secreted miR-153 is released into exosomal or microvesicle structures, we isolated by differential centrifugation exosomal and microvesicles fractions from CM of DU145 and PC3 and analyzed the expression of miR-153 in each fraction. Both fractions and cell homogenates were tested for the presence of the Tumor Susceptibility Gene 101 (TSG101) protein, a typical exosomal marker, by Western blot analysis. As shown in Figure 9A, TSG101 is expressed by DU145 and PC3 cells and is specifically present only in the 100,000× *g* fraction (Ex), indicating a proper recovery of exosomes and the absence of Ex contamination in the MV fraction (16,000× *g*). RT-qPCR analysis revealed that miR-153 is secreted in both exosome and microvesicles from DU145, with a higher fraction of miR-153 in exosomes; instead, miR-153 is present mainly in the microvesicles of PC3 cell CM (Figure 9B,C, respectively).

## 3. Discussion

MicroRNAs are merging as key molecules in the regulation of several physiological and pathological processes. Starting from a previous work, in which we demonstrated using in silico analysis the importance of miR-153 in the diagnosis of higher Gleason score PC, we tried to understand the molecular role of this miRNA specifically in high GS PC cell lines, with DU145 and PC3 cells corresponding respectively to GS7 and GS8. In particular, we confirmed that miR-153 is highly expressed in cell lines and primary tissues with high GS levels, as already suggested by [13]. In our experiments, the silencing of endogenous miR-153 by As oligonucleotide treatment revealed that this miRNA is involved in the control of cell proliferation of both PC3 and DU145 cell lines. As suggested by the luciferase assay results, this effect could be mediated by the direct regulation of miR-153 on its target *KLF5*, whose expression increased when we silenced miR-153. *KLF5* protein is a zinc-finger transcription factor involved in the control of proliferation, migration and differentiation of different types of tumors [16,17,18]. It is already known that *KLF5* is a key controller of cell proliferation, as in bladder cancer, where it regulates cyclin D1 or Cyclin E expression [19,20]. In addition, in PC, *KLF5* controls proliferation, possibly interacting with androgen receptor (AR) and activating Myc transcription [21].

In DU145 and PC3, miR-153 controls invasion and migration, as suggested by the results of the wound healing and Boyden’s chambers experiments. This suggests that the presence of miR-153 is necessary to maintain the mobility of the cells, associated with an aggressive phenotype. In breast cancer, miR-153 has a known role as a controller of migration, invasion and the process of epithelia-to-mesenchymal transition [22], but miR-153 expression in this tumor is opposite compared to prostate cancer. In hepatocellular carcinoma, miR-153 is also downregulated and inhibits migration by controlling epithelial to mesenchymal transition (EMT) and targeting the expression of SLUG [23].

The results of the experiment performed with conditional medium suggest that DU145 and PC3 cells, overexpressing miR-153 in respect to BPH1, could secrete higher amounts of this miRNA in the surrounding environment; secreted miR-153 could enter the surrounding cells, modulating the expression of *KLF5* mRNA. In prostate cancer cells, the overexpression of miR-153 is necessary to maintain *KLF5* silencing, which in turn maintains a high cell proliferation rate. On the contrary, the exogenous increase in miR-153, from CM of DU145 or PC3, in non-tumoral BPH1 cells, although regulating *KLF5*, behaves as a tumor suppressor gene, decreasing the proliferation rate of the cells. These opposite behaviors could be due to a dual role that several genes have, depending on the context in which they are expressed, as already demonstrated by [24]. miR-153 also has a dual role in different cancer types, being an oncogene in colorectal cancer [25] or a tumor suppressor in gastric cancer [26]. It is thus possible that miRNAs have a functional dualism, depending on the tissue context in which their expression is altered.

We also observed that miR-153 could be secreted by higher GS cell lines in different compartments, being differentially distributed between exosomes and microvesicles. In PC, it is common to find secretion of miRNAs, both in urine [27] and semen liquid [28], but also in blood. The body fluids can be a useful source for miRNA isolation and analyses, and several circulating miRNA profiles have been proposed for the precise stratification of PC (i.e., [27]), alone or in combination with PSA marker [28]. In addition, miRNA profiles containing a combination of miRNA molecules could be helpful in the identification of PC and of metastatic forms of PC. In 2019, the combination of PSA, blood miR-21 and miR-221 expression and the FDA-approved blood test Prostate Health Index (PHI) were proposed as a diagnostic and prognostic tool for PC [29]. Considering the need of developing cheaper and minimally invasive biomarkers for early differential diagnosis of PC, the evaluation of miR-153 expression levels in blood could be an advancement in the clinical approach for PC diagnosis, also supported by the correlation between tissue miR-153 overexpression and PC poor prognosis [13].

Secreted miRNAs have been proposed as non-proteic molecules for cell–cell communication. Secreted miR-153 could enter into the surrounding tissues, altering the behavior of neighborhood cells; we have demonstrated that miR-153 could be overexpressed in BPH1, usually having lower miR-153 compared to PC3 and DU145, when grown into CM from DU145 and PC3. In BPH1 grown in the conditioned medium of both PC cell lines, the increase in miR-153 impacts the expression of endogenous *KLF5*. The entrance of miR-153 in BPH1 impacts cell proliferation by decreasing the growth rate of the cells.

## 4. Materials and Methods

### 4.1. Cell Lines

PC3 (ATCC cod.CRL-1435, Androgen negative grade IV adenocarcinoma), Gleason Score (GS) 8, and DU145 (ATCC cod.HTB-81), GS 7, cell lines were all acquired from the American Type Culture Collection (ATCC, Manassas, VA, USA). PC3 cells are a grade IV prostate adenocarcinoma. Gleason Grading system and the recent changes from International Society of Urological Pathology consensus conference on Gleason Grading of Prostatic Adenocarcinoma suggested that grade IV corresponds to Gleason score 8 [30]. DU145 are aggressive invasive adenocarcinoma from metastatic lesion. They are reported to be a grade II adenocarcinoma (by ATCC website), which corresponds to Gleason score 7 [30]. BPH1 benign human epithelial hyperplastic immortalized cell line was acquired from Interlab Cell Line Collection (ICLC) (Genova, Italy). DU145 and PC3 cells were cultured in RPMI, containing 10% heat-inactivated fetal bovine serum (FBS) (Euroclone, Pero, Italy), 100 U/mL penicillin (Euroclone), 100 μg/mL streptomycin (Euroclone), and 2 mM L-glutamine (Euroclone,), 100 mM sodium pyruvate (Euroclone). BPH1 cells were cultured in RPMI supplemented with 20% FBS (Euroclone), 2 mM l-Glutamine (Euroclone), and 100 U/mL penicillin (Euroclone), 100 μg/mL streptomycin (Euroclone). All cell lines were maintained in a humidified 37 °C incubator with constant 5% CO_2_ and split for maintenance at 60–70% confluency.

### 4.2. Oligonucleotide Cell Treatment

In order to obtain the decrease in expression of miR-153-2-3p in PC cell lines, we treated the cells with 200 nM of synthetic antisense (As) oligonucleotide, with the following sequence 5′-GATCACTTTTGTGACTATGCAA-3′, which is the reverse and complementary sequence of miR-153-2-3p (miRbase Accession number MI0000464). As a comparison treatment, the cells were grown in the presence of a scrambled (Scr) oligonucleotide (5′-ATGATGTCCTTCTAGTACGCATC-3′). The oligonucleotides were melted in the growth medium of the cells, without any transfection agent. The cells were harvested at least after 48 h, as indicated in each experiment. The concentrations of the As miR-153-2-3p (As miR-153) and Scr oligonucleotides were chosen on the basis of the results of previous experiments on the effect of different dosage of oligonucleotides on the expression of miRNA (our experiments, not shown). All the primers have been provided by IDT (Tema Ricerca, Bologna, Italy).

### 4.3. UALCAN Analysis

To calculate the distribution of miR-153-2-3p expression in PCa patient, we used UALCAN software (http://ualcan.path.uab.edu/index.html (accessed on 5 June 2021)) [14]. This software is a user-friendly and interactive web source for the analysis of OMICS data. This software allows for the analysis of the expression levels of miR-153-2-3p in PCa tumor tissue from TCGA database, which includes 497 prostate cancer samples (adenomas, adenocarcinomas, ductal and lobular neoplasms, cystic, mucinous and serous neoplasms) and 52 normal samples.

### 4.4. Luciferase Reporter Assay

The 150 bp fragment of *KLF5* 3′UTR containing the seed for miR-153 binding (74 bp upstream and 68 bp downstream the seed) was amplified by Fw primers containing EcoRI site (GAATTCGCTATGcacctacatgaaaag) and the Rw primer containing the XhoI site (5′-CTCGAGGCTATGgtcgttaatatataaacatc). All the primers have been provided by IDT (Tema Ricerca, Bologna, Italy). The fragment was cloned into pEZX-MT05 vector (GeneCopeia, Tebu-Bio, Le Perray, France) using the HmiT111619-MT05 clone construct, which is the pEZX-MT05 vector (GeneCopeia) in which the 3′UTR of *CXCR4* was cloned by EcoRI/XhoI sites (GeneCopeia). In order to generate the construct containing 3′UTR of *KLF5*, we cut out the fragment of the 3′UTR of *CXCR4* by digestion with EcoRI and XhoI enzymes (New England Biolabs, Hitchin, UK) and cloned in the 3′UTR of *KLF5* by EcoRI/XhoI cloning sites. The empty construct (in which the blunt ends obtained by digestion of HmiT111619-MT05 clone by EcoRI/XhoI and Klenow-treatments where ligated) was used as a negative control. The vectors were transfected by Lipofectamine (Invitrogen, Thermo Fisher Scientific, Waltham, MA, USA) into PC3 and DU145 cell lines in combination with As miR-153, S miR-153 or Scrambled oligonucleotide. Twenty-four hours after transfection, the luciferase assay was performed (Secrete Pair Dual Luminescence kit, Genecopeia, distributed by Tebu-Bio, Le Perray, France), normalizing the expression over Shrimp Alkaline Phosphatase (SEAP) activity.

### 4.5. MTT Assays

Methylthiazol tetrazolium (MTT) assays were performed to determine the cell line’s sensitivity to oligonucleotide treatment. Then, 5000 cells per well were plated in 96-well plates in 100 μL of complete medium and were allowed to attach overnight under normal culture conditions. The next day, the cells were treated with 200 nM As or Scr oligonucleotide of miR-153-2-3p in 100 μl of medium. Cells were stained with 10 μL of 5 mg/mL MTT solution (SigmaAldrich, Burlington, MA, USA) for 4 h at 37 °C, at 24, 48, 72 h after the treatment. Media were then removed, and 100 μl of dimethyl sulfoxide (DMSO) was added to resuspend formazin crystals 30 min at room temperature. Plates were read on a Fluor Star Omega plate photometer (Euroclone, Siziano, Italy) at a wavelength of 570 nm. Colorimetric readings were normalized against plates of non-treated cells under identical culture conditions. The experiment was performed in triplicate. Statistical *t* test was applied to evaluate differences in cell proliferation (*p* ≤ 0.05 was considered significant).

### 4.6. Wound Healing and Boyden’s Chamber

PC3 and DU145 were seeded in 24-well plates at full confluency (150,000 cells/w). The day after seeding, the wound was drawn, the medium was removed, and the new medium containing Scr or As miR-153 oligonucleotide was added. The images were taken at 0 and 24 h after the beginning of the treatment. Fiji public domain software (NIH, Bethesda, MD, USA) was used to quantify the migration of the cells, as area of the wound in each condition at time 0 h and 24 h. The calculation of wound closure % was performed following the paper of Grada [31].

In order to understand the role of miR-153 in invasion ability, we performed a Boyden’s chamber assay, as described in [32].

### 4.7. Conditional Medium Treatment and Circulating miRNA Isolation

PC3 and DU145 cells were cultured at 70% confluency in complete RPMI medium. After 48 h, the conditional medium was collected, centrifuged to remove cellular debris and stored at 4 °C until used. The conditional medium was then placed onto BPH1 cells at 70% confluency, for 24, 48, 72 h. The cells were harvested and collected for RNA extraction and RT-qPCR analysis. miRNA isolation on conditional medium was perform with miRNeasy serum/plasma kit (Qiagen, Milan, Italy), following manufacturer’s instructions. The isolated RNA was used for retrotranscription and RT-qPCR analyses.

### 4.8. Cell Count

In order to understand whether secreted miR-153 in the conditional medium affect the proliferation rate of the BPH1 cells, we seeded the cells in 6-well plates (100,000/well). As soon as the cells attached (4 h), we added the conditional medium over the cells and counted the cells every 24 h.

### 4.9. Exosomes (Ex) and Microvesicles (MV) Isolation by Differential Ultracentrifugation

Conditioned medium of DU145 and PC3 cells was harvested after 48 h and centrifuged at 300× *g* for 10 min at 4 °C to pellet dead cells and bulky debris. The obtained supernatant was then centrifuged at 16,000× *g* for 20 min at 4 °C in order to collect the MV fraction. The MV pellet was washed once using PBS at 16,000× *g* for 20 min at 4 °C and subjected to RNA extraction. The remaining supernatant was filtered through a 0.22 µm filter (Merck Millipore, Burlington, MA, USA) to remove contaminating apoptotic bodies, residual microvesicles and cell debris and subsequently subjected to 100,000× *g* centrifugation in a Beckman Coulter Optima™ L-90K Ultracentrifuge (BC, Brea, CA, USA) with a Type 60 Ti rotor for 70 min at 4 °C. The 100,000× *g* pellet was resuspended in 1 mL of ice-cold PBS and subjected to a second round of centrifugation at 100,000× *g* for 60 min at 4 °C, in a Beckman TL100 ultracentrifuge (BC, Brea, CA, USA) with TLA 100.3 fixed angle rotor. The resulting exosome pellet was collected and subjected to RNA extraction.

### 4.10. SDS-PAGE and Western Blot

PC3 and DU145 cells were washed 2 times with PBS pH 7.4 at 4 °C and mechanically harvested in lysis buffer (20 mM Tris-HCl pH 7.4, 150 mM NaCl, 1% NP40, 10 mM NaF, 1 mM Na_3_VO_4_, 10 mM sodium pyrophosphate, 2 µg/mL Pepstatin, 2 µg/mL Aprotinin, 2 µg/mL Leupeptin, 1 mM PMSF; all these reagents are from SigmaAldrich, Burlington, MA, USA). Samples were then mixed for 20 min at 4 °C and centrifuged at 11,400× *g* for 10 min at 4 °C. Following supernatant separation, Sample Buffer 4X was added and samples were boiled at 100 °C for 5 min and then stored at −20 °C.

Pellets of 16,000× *g* and 100,000× *g* fractions, obtained from DU145 and PC3 conditioned media, were dissolved in 50 μL of ice-cold PBS pH 7.4 and mixed with Sample Buffer 4X, boiled at 100 °C for 5 min and stored at −20 °C.

All samples were then separated by 10% SDS-PAGE gel and transferred to nitrocellulose membrane. Membrane was blocked with 5% skim milk in Tris Buffered Saline with 0.1% Tween 20 (TBST) and then incubated overnight at 4 °C with anti TSG101 antibody (Immunological Sciences, Rome, Italy) 1:1000 in TBST with 5% skim milk. After washing membrane with TBST four times, membrane was incubated with horseradish peroxidase (HRP)-conjugated secondary antibody in TBST with 5% skim milk. After washing, protein bands were visualized using ECL system (Cyanagen, Bologna, Italy).

### 4.11. RNA Extraction and RT-qPCR Analysis

RNA extraction was performed by the Trizol method (Invitrogen, Thermo Fisher Scientific, Waltham, MA, USA), following manufacturer’s instructions. After extraction, RNA was quantified by Bio Photometer (Eppendorff, Enfield, CT, USA), and 1 μg of total RNA was reverse transcribed after DNAse treatment (both Thermo Fisher Scientific). Sybr Green-based (BioRad, Hercules, CA, USA) real time-quantitative PCR (RT-qPCR) was performed in CFX connect Real Time System (BioRad, Segrate, Italy) using the following homemade primers (Integrated DNA Technology -IDT-, Tema Ricerca, Bologna, Italy): *14S* Fw primer 5′-GGCAGACCGAGATGAATCCTCA-3′ Rw primer 5′-CAGGTCCAGGGGTCTTGGTCC-3′, *KLF5* Fw primer 5′-AATTTACCCACCACCCTGCC-3′ Rw primer 5′-TGTGCAACCAGGGTAATCGC-3′; *BAX* Fw primer 5′-CCTGTGCACCAAGGTGCCGGAACT-3′, Rw primer 5′-CCACCCTGGTCTTGGATCCAGCCC-3′; *BCL2* Fw primer 5′-GATTGTGGCCTTCTTTGAG-3′, Rw primer 5′-CAAACTGAGCAGAGTCTTC-3′; *BAD* Fw primer 5′-CCCAGAGTTTGAGCCGAGTG-3′, Rw primer 5′-CCCATCCCTTCGTCGTCCT-3′.

For miRNA reverse transcription, the MystiCq microRNA cDNA synthesis kit (MIRRT) was used (Sigma Aldrich, Burlington, MA, USA). The primer used for miR-153 amplification was designed by OligoAnalyzer IDT tool (Integrated DNA Technology, IDT): miR-153 Fw primer 5′-TTGCATAGTCACAAAAGTGATC-3′. As a housekeeping gene, the internal control of the kit was used.

### 4.12. Analysis of miR-153 Secretion in Exosomes and Microvesicles

In order to quantify the miR-153 in exosome and microvesicles, we first collected the 48 h conditional medium from 70% confluent cells, both for PC3 and DU145. Then, the exosomes and microvesicles were isolated by differential ultracentrifugation, as described above. The RNA was extracted, and the analysis of miR-153 was performed by RT-qPCR using the MystiCq microRNA cDNA synthesis kit (MIRRT) (Sigma Aldrich, Burlington, MA, USA). The level of expression of miR-153 in each sample was normalized on an internal control of the kit. PC3/DU145 exosomal and microvesicle miR-153 expression was compared to that of cellular PC3 or DU145.

## 5. Conclusions

miR-153 controls high GS prostate cancer cell proliferation, migration and invasion, possibly by regulating *KLF5* expression. Moreover, miR-153 is secreted in PC cell culture fluids, both by microvesicles and exosomes, influencing the behavior of surrounding tissues. This miRNA, analyzed in the blood of PC patients, could become a diagnostic molecule for higher Gleason score PC.

## Figures and Tables

**Figure 1 ijms-23-06339-f001:**
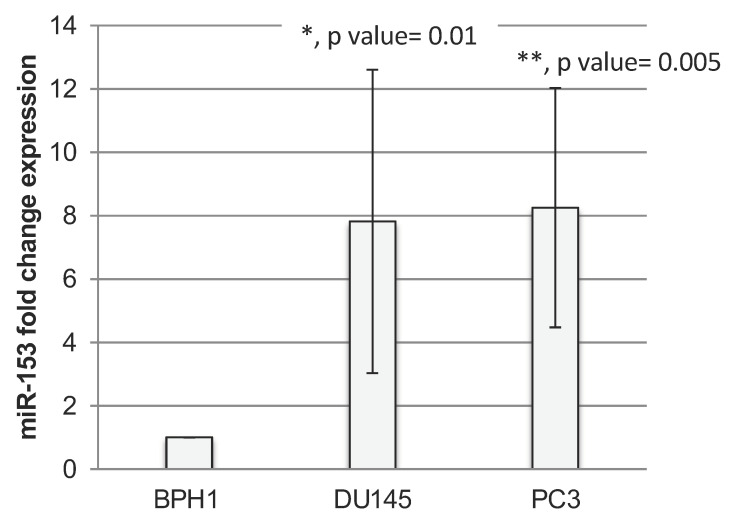
**miR-153 is overexpressed in prostate cancer cell lines.** RT-qPCR fold change expression (2^−ΔΔCt^ ± standard deviation-sd-values) of miR-153 in prostate cancer DU145 and PC3 cell lines. Summary of two experiments in triplicate (*t* test, *p* value ≤ 0.05, *; 0.01, **).

**Figure 2 ijms-23-06339-f002:**
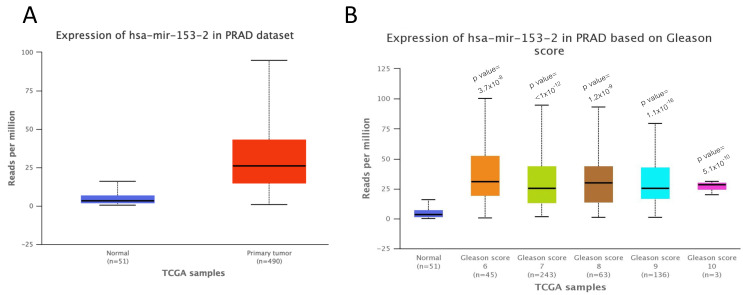
**miR-153 overexpression in prostate adenocarcinoma is associated with higher Gleason score and poor prognosis.** (**A**). UALCAN analysis of the expression levels of miR-153 in prostate adenocarcinoma (PRAD) primary tumors (*n* = 490) compared to normal samples (*n* = 51) (*p* value = 1.62 × 10^−12^). (**B**). Expression of miR-153 in different samples from TCGA, based on their Gleason score (*p* values have been calculated between PC and normal tissue samples).

**Figure 3 ijms-23-06339-f003:**
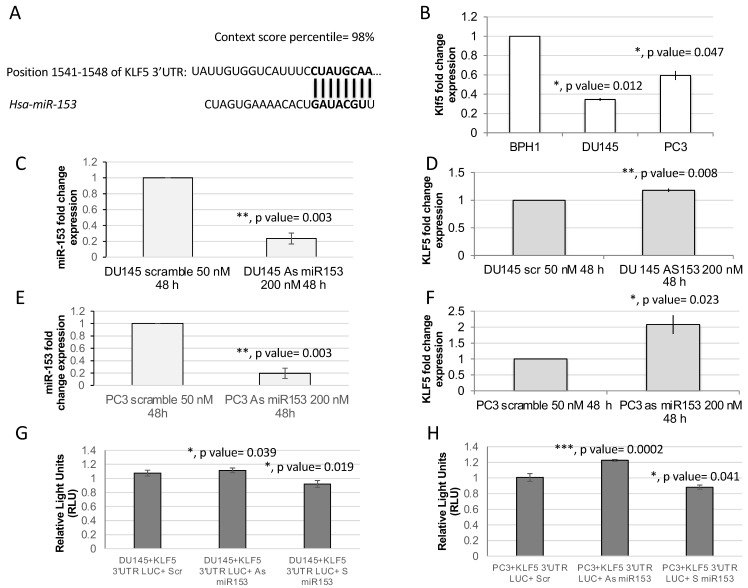
***KLF5* is a putative target of miR-153.** (**A**) Targetscan prediction revealed a single 8-mer highly conserved binding site of miR-153 on *KLF5* (**A**) and two less conserved binding sites. (**B**) RT-qPCR analysis revealed in DU145 and PC3 that there is a downregulation of *KLF5* mRNA expression (fold change expressed as 2^−ΔΔCT^ ± sd values are shown; average of two independent experiments in triplicate; *t* test, *p* value ≤ 0.05, *; 0.01, **; 0.001, ***). (**C**,**E**). Modulation of miR-153 expression in DU145 (panel **C**) and in PC3 (panel **E**) calculated by RT-qPCR as fold change expression (2^−ΔΔCT^ ± sd values) compared to scrambled-treated cells. (**D**,**F**) *KLF5* expression in DU145 (panel **D**) and in PC3 (panel **F**) calculated by RT-qPCR as fold change expression (2^−ΔΔCT^ ± sd values) compared to scrambled-treated cells. (**G**,**H**) Luciferase assay performed in DU145 (**G**) or in PC3 (**H**) cells in the presence of Scr, As miR-153 or S miR-153 oligonucleotides. *t* Test *p* values are indicated (calculated versus scr-treated cells).

**Figure 4 ijms-23-06339-f004:**
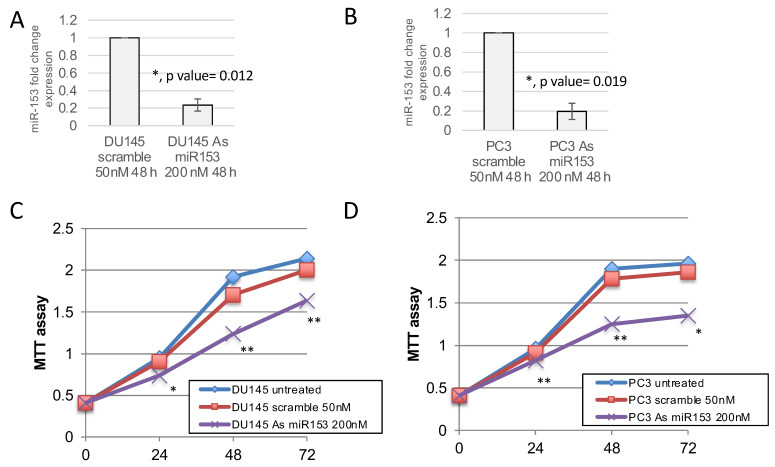
**miR-153 downregulation decreases the growth of prostate cancer cell lines.** MTT assay revealed that the modulation of miR-153 expression, obtained by 200 nM of antisense (As) oligonucleotide treatment and revealed by RTqPCR (fold change expression, as 2^−ΔΔCT^ ± sd values), in DU145 (**A**) and PC3 (**B**) cell lines decreased cell growth in both cell lines (**C**,**D**), respectively). Summary of two experiments in triplicate (*t* test compared to scramble treated cells, *p* value ≤ 0.05, *; 0.01, **).

**Figure 5 ijms-23-06339-f005:**
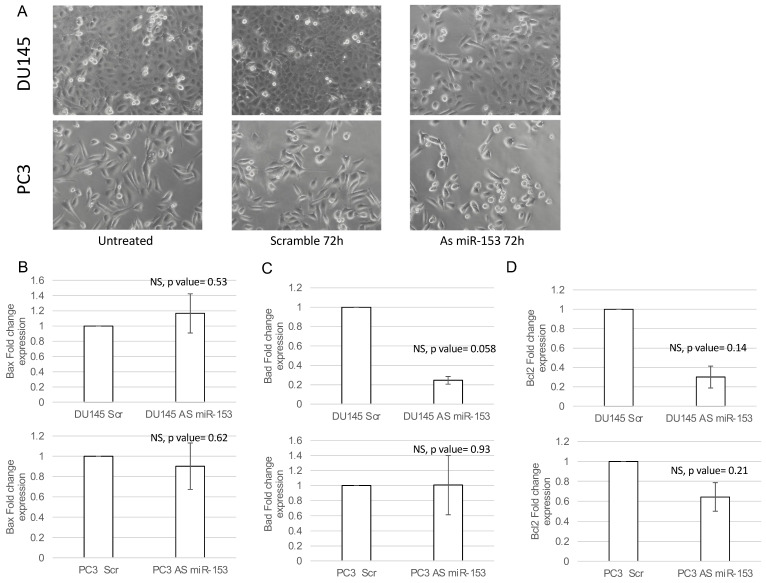
**miR-153 silencing caused decreased confluency in both cell lines.** (**A**) Phase contrast 20× images of DU145 and PC3 untreated (on the left), 200 nM Scr-treated (on the center), or As miR-153-treated cells (on the right). Fold change expression (2^−ΔΔCT^ ± sd) of Bax (**B**), Bad (**C**) and Bcl2 (**D**) apoptotic genes in DU145 (upper panels) or PC3 (lower panels) treated with 200 nM 72 h scramble (Scr) or Antisense miR-153 (As) oligonucleotides. Summary of two experiments in triplicate (*t* test compared to scramble treated cells, NS, not significant).

**Figure 6 ijms-23-06339-f006:**
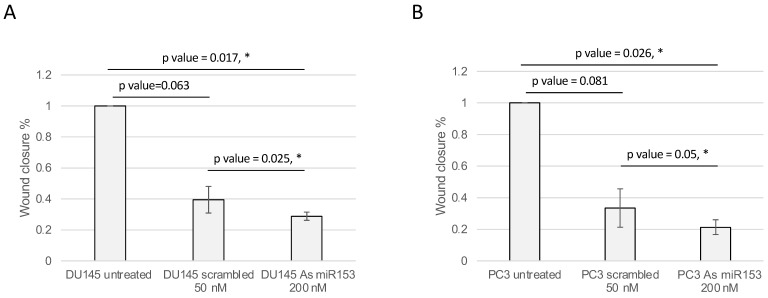
**As miR-153 reduced the migration ability of both DU145 and PC3 cells.** Calculation of the wound closure % in DU145 (**A**) and PC3 (**B**) cells treated with scrambled or As miR-153 oligonucleotides. Wound healing is calculated as: (area of the wound T = 24 h-area of the wound T = 0)/area of the wound T = 0. The treatment with As miR153 decreased the ability of the cell to close the wound, both compared to Scr-treated cells or to untreated cells. *t* Test *p* values are indicated (*n* = 3 images for each sample in each time point; experiment performed in duplicate) (*t* test, *p* value ≤ 0.05, *).

**Figure 7 ijms-23-06339-f007:**
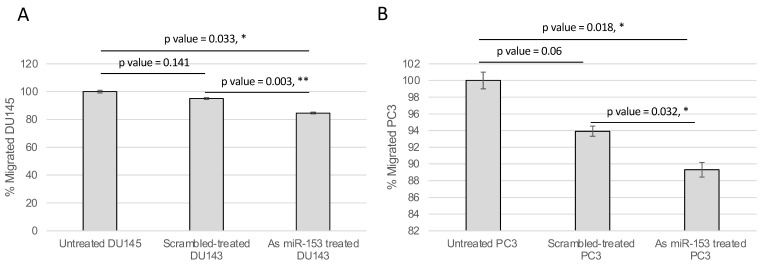
**miR-153 regulates migration of PC cells.** The migration ability of the cells through the Boyden’s chamber was significantly affected by As miR-153 treatment, compared to both untreated and scrambled-treated cells, in DU145 (**A**) and PC3 (**B**) cells. *t* Test *p* values are indicated (quantification of migrated cells in *n* = 4 images for each sample at 24 h of treatment; experiment performed in duplicate) (*t* test, *p* value ≤ 0.05, *; 0.01, **).

**Figure 8 ijms-23-06339-f008:**
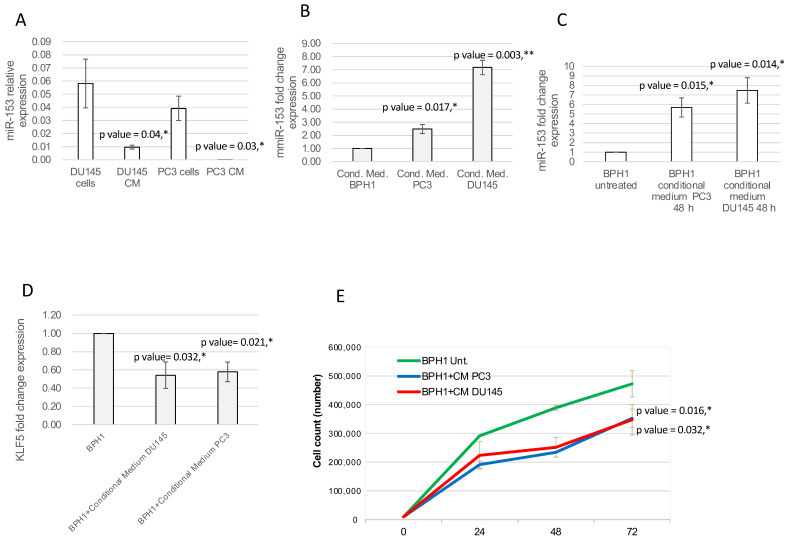
**Secreted miR-153 can enter in the surrounding cells.** (**A**) DU145 and PC3 can secrete miR-153 in the conditional medium. RT-qPCR analysis (relative expression, 2^−ΔΔCT^ ± sd) of miR-153 and internal control was performed on conditional medium collected from DU145 and PC3 cell lines (*t* test, *p* value ≤ 0.05, *). (**B**) miR-153 is highly expressed in conditional medium from PC3 and DU153 cells compared to BPH1 (miR-153 fold change expression, as 2^−ΔΔCT^ ± sd values, is shown; *t* test, *p* value ≤ 0.05, *; 0.01, **). (**C**) DU143 and PC3 conditional medium is able to enter into BPH1 cells, significantly increasing its inner level of expression (miR-153 fold change expression, as 2^−ΔΔCT^ ± sd values, is shown; *t* test, *p* value ≤ 0.05, *). (**D**) *KLF5* fold change expression (2^−ΔΔCT^ ± sd values) is significantly affected by internalized miR-153. *t* test *p* values compared to untreated BPH1 cells are indicated (*n* = 2 experiments performed in triplicate; *t* test, *p* value ≤ 0.05, *). (**E**) Count of BPH1 cells in standard medium or in conditional medium (CM) from DU145 or PC3. *t* test *p* values compared to untreated BPH1 cells are indicated (*n* = 2 experiments performed in triplicate; *t* test, *p* value ≤ 0.05, *).

**Figure 9 ijms-23-06339-f009:**
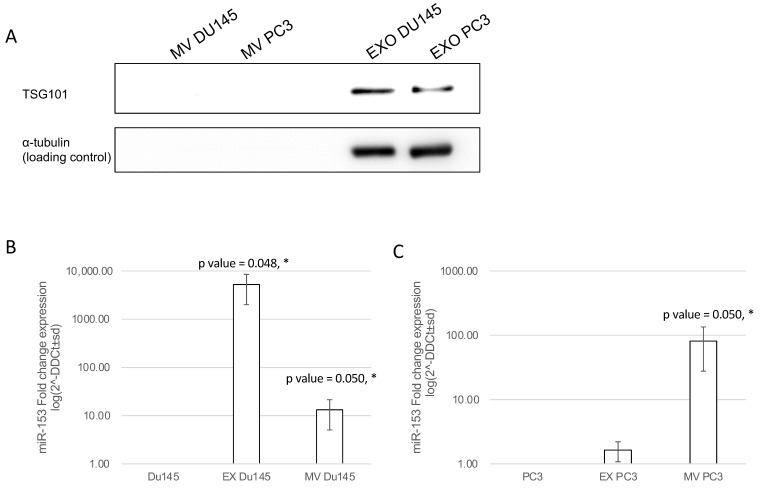
**miR-153 is secreted by microvesicles (MV) and exosomes (Ex).** (**A**) DU145 and PC3 cell homogenates together with microvesicles and exosome fractions obtained from DU145 and PC3 conditioned media were analyzed by Western blot for the exosome marker TSG101. Then, 30 μg of cell homogenate and the corresponding MV and Ex isolated from 1.5 mg of cell proteins were loaded. TSG101 protein was found in cells and in the exosome fraction. MV and Ex isolated from conditional medium of PC3 (**B**) and DU145 (**C**) cells showed a significant increase in miR-153 expression, compared to the content of intracellular miR-153 of DU145 or PC3 cells, used as reference. Log scale of fold change expression is represented. Average of two independent experiments performed in duplicate (*t* test, *p* value ≤ 0.05, *).

## Data Availability

Not applicable.

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
