# Peer review of "Secreted miR-153 Controls Proliferation and Invasion of Higher Gleason Score Prostate Cancer"

_ijms, 2022, doi:10.3390/ijms23116339_

Round 1

Reviewer 1 Report

In the first part of the paper Authors presented data to demonstrate that miR-153, a miRNA already associated with poor prognosis in prostate cancer (PCa), i) is upregulated in PCa primary tumor and increase in tumor with the increase of Gleason Score (GS) according to TCGA database and ii) increased cell proliferation and migration/invasion in vitro by targeting KFL5. In the second part of the paper Authors try to demonstrate that miR-153 secreted from high GS PCa cells increase cell proliferation of benign BPH1 cells by targeting KFL5 as well.

The study presents several flaws and crucial data are not convincing or do not support the conclusions.

 Broad comments

1) The oncogenic role of miR-153 in PCa is not completely new as Authors state in the Introduction, as Wu et al. (2013, PMID 23060044) already demonstrated that the miR-153 overexpression increases DU-145 and 22Rv1 proliferation by direct targeting PTEN.

2) According to the text (since that the Figure 1C is missing) the UALCAN software-based analysis reveals the association between “high miR-153 expression/poor prognosis” but not in a significant way. Given that this association has already been demonstrated, this result raises some doubts about the reliability of UALCAN software. In Figure 1B the significance is not indicated.

3) The Authors demonstrated that miR-153 silencing results in a decrease of cell proliferation and an increase KFL5 and conclude that KFL5 could be a possible miR-153 target. If the Authors demonstrated (with a gene reporter assay) that KFL5 is a direct target of miR-153 the manuscript would benefit. I have some doubts about the “miR-153 overexpressing method” without any transfectants. The Authors do not explain why they needed to use different “antisense concentration” (among scramble and miR-153 samples) and how they decide the right concentrations. Finally, I think that the data regarding apoptosis should be shown.

4) Authors state that PC3/DU-145 cells release miR-153 and that the secreted miR-153 enter BPHI cells and downregulates KFL5 increasing cell proliferation. First of all, Authors must demonstrate that BPHI cells do not express miR-153 or indicate a reference in which it is demonstrated. Moreover, they must demonstrate that the level of miR-153 in the PC3/DU-145 medium is higher in respect of BPHI medium. Secondly and more important, data reported in figure 8C do not indicate that the proliferation of BPHI cells treated with PC3/DU-145 media increase in respect of the untreated ones as the Authors said but exactly the contrary.

5) I have several concerns regarding miR-153 quantification in the exosomes (Ex) and macrovesicles (Mv). The qRT-PCR data in Figure 9 A and B reported a fold change increase of miR-153 in “PC3/DU-145 Ex” and “PC3/DU-145 Mv” samples compared to “DU-145” and “PC3” samples. What are “DU-145” and “PC3” samples (that Authors use as “reference”)? Authors must better explain how they performed the experiment (e.g. which internal control they used to normalize the data and whether they used a spike-in oligo) because the analysis could be affected by several bias that undermine the credibility of the results.

Author Response

Comments and Suggestions for Authors

In the first part of the paper Authors presented data to demonstrate that miR-153, a miRNA already associated with poor prognosis in prostate cancer (PCa), i) is upregulated in PCa primary tumor and increase in tumor with the increase of Gleason Score (GS) according to TCGA database and ii) increased cell proliferation and migration/invasion in vitro by targeting KFL5. In the second part of the paper Authors try to demonstrate that miR-153 secreted from high GS PCa cells increase cell proliferation of benign BPH1 cells by targeting KFL5 as well.

The study presents several flaws and crucial data are not convincing or do not support the conclusions.

 Broad comments

  • The oncogenic role of miR-153 in PCa is not completely new as Authors state in the Introduction, as Wu et al. (2013, PMID 23060044) already demonstrated that the miR-153 overexpression increases DU-145 and 22Rv1 proliferation by direct targeting PTEN.

We thank the reviewer for this observation. We think that until now no publication explains the functional, biological role of miR-153 in PC with high GS (GS≥7). This is the reason why we focused on this specific miRNA, to explain its role in PC with an estimate poor prognosis. Moreover, the article of Wu et al. (The prostate, 2012) describing the regulatory role of miR-153 on PTEN and on the pathway of FOXO and AKT kinase does not describe the possibility that miR-153 is secreted into the cell culture medium.

  • According to the text (since that the Figure 1C is missing) the UALCAN software-based analysis reveals the association between “high miR-153 expression/poor prognosis” but not in a significant way. Given that this association has already been demonstrated, this result raises some doubts about the reliability of UALCAN software. In Figure 1B the significance is not indicated.

We thank the reviewer for these observations. We corrected the text according to the absence of figure 2C. We also indicated, as suggested, the p values in figure 2B of the revised manuscript (section ‘2.Results’, line 104-108 and new figure 2B line 109 of the revised manuscript).

We accept the opinion of the referee about the reliability of UALCAN software, but we believe in the reproducibility of the software as presented in previous works [i.e., Zhang Y, Chen F, Chandrashekar DS, Varambally S, Creighton CJ. Proteogenomic characterization of 2002 human cancers reveals pan-cancer molecular subtypes and associated pathways. Nat Commun. 2022 May 13;13(1):2669. doi: 10.1038/s41467-022-30342-3] [Chandrashekar DS, Karthikeyan SK, Korla PK, Patel H, Shovon AR, Athar M, Netto GJ, Qin ZS, Kumar S, Manne U, Creighton CJ, Varambally S. UALCAN: An update to the integrated cancer data analysis platform. Neoplasia. 2022 Mar;25:18-27. doi: 10.1016/j.neo.2022.01.001].

We could explain the non-significant association between “high miR-153 expression/poor prognosis” because UALCAN software is based on third quartile of miR-153 for the high-expression group. Thus, the limitation of UALCAN software could be the inability of the user   to choose an optimal cut-off specific for his own data.

We have included the limitation of UALCAN software in the revised manuscript line104-108.

  • The Authors demonstrated that miR-153 silencing results in a decrease of cell proliferation and an increase KFL5 and conclude that KFL5 could be a possible miR-153 target. If the Authors demonstrated (with a gene reporter assay) that KFL5 is a direct target of miR-153 the manuscript would benefit.

We thank the reviewer for the suggestion. We cloned the 3’UTR of KLF5 target gene into pEZX-MT05 vector and performed the Luciferase assay, as described in section Material and Methods, par.4.4 Luciferase Reporter Assay (lines 349-360) of the revised manuscript. The results are presented in the new Fig.3 (panel G and H), lines 136-137 and 146-148 of the par.2.2 in the Results section; lines 267-268 of the Discussion section in the revised manuscript.

I have some doubts about the “miR-153 overexpressing method” without any transfectants.

We thank the reviewer for this observation. Indeed, we have already succeeded in overexpressing a cluster of miRNAs in triple negative breast cancer cells by just adding the oligonucleotides in the medium (as reported in doi: 10.1093/nar/gkz016). In other projects (i.e doi: 10.1038/s41598-021-85746-w) we needed the presence of a transfectant reagent in order to obtain the expression of the miRNA. Indeed, in the case of miR-153, we tested the need or not of a transfectant agent before starting the project. We don’t know the reason of these differences; is it possible that the sequence of the miRNA oligonucleotide or the charge of the bases within the sequence influence the entrance in the cells?

The Authors do not explain why they needed to use different “antisense concentration” (among scramble and miR-153 samples) and how they decide the right concentrations. 

We thank the reviewer for this observation. We highlighted in the manuscript that the concentrations of the oligonucleotides have been chosen on the base of toxic cellular response in previous titration experiments in Material and Methods section (Par.’4.2 Oligonucleotide cell treatment’, lines 337-340 of the revised manuscript).

Finally, I think that the data regarding apoptosis should be shown.

We thank the reviewer for this observation. We inserted the results on the apoptosis in Fig.5B-D (line 158, 169-170, and 173-176) of the revised manuscript. The sequences of the primers were added in Material and Methods, 4.11 RNA extraction and RT-qPCR analysis subsection (line 441-445 of the revised manuscript).

  • Authors state that PC3/DU-145 cells release miR-153 and that the secreted miR-153 enter BPHI cells and downregulates KFL5 increasing cell proliferation. First of all, Authors must demonstrate that BPHI cells do not express miR-153 or indicate a reference in which it is demonstrated.

The reviewer is right. BPH1 cell line expresses lower level of miR-153 compared to PC3 or DU145. We corrected the sentence, as reported in line 212 Section ‘2.5 miR-153 could be secreted, influencing surrounding tissue’ and also in line 304, section ‘3. Discussion’ of the revised manuscript.

Moreover, they must demonstrate that the level of miR-153 in the PC3/DU-145 medium is higher in respect of BPHI medium. 

We thank the reviewer for the question. We isolated and amplified miR-153 and internal control in conditional medium of both cell lines, the results are presented in the revised figure 8A, line 225-226 (the relative expression of miR-153 in conditional medium compared to cellular content for both cell lines); in Results (2.5 miR-153 could be secreted, influencing surrounding tissue, lines 207-211), in the new legend of figure 8 (lines 228-229), in Material and methods (section 4.7 Conditional medium treatment and circulating miRNA isolation, lines 391-393) and were modified accordingly in the revised manuscript.

Secondly and more important, data reported in figure 8C do not indicate that the proliferation of BPHI cells treated with PC3/DU-145 media increase in respect of the untreated ones as the Authors said but exactly the contrary.

The reviewer is right. We corrected the text accordingly (line 224, Section ‘2.5 miR-153 could be secreted, influencing surrounding tissue.’ of the revised manuscript).

  • I have several concerns regarding miR-153 quantification in the exosomes (Ex) and macrovesicles (Mv). The qRT-PCR data in Figure 9 A and B reported a fold change increase of miR-153 in “PC3/DU-145 Ex” and “PC3/DU-145 Mv” samples compared to “DU-145” and “PC3” samples. What are “DU-145” and “PC3” samples (that Authors use as “reference”)?

Thanks for this observation. The miR-153 expression in exosomes or microvesicles has been compared to the content of intracellular miR-153 of DU145 and PC3 cells, respectively. We clarify this point in Legend of Figure 9 (line 255-256) of the revised manuscript.

Authors must better explain how they performed the experiment (e.g. which internal control they used to normalize the data and whether they used a spike-in oligo) because the analysis could be affected by several bias that undermine the credibility of the results.

We thank the reviewer for this observation. Starting from the same amount of conditional medium for each cell line, the exosomes and microvescicles have been isolated, as described in section ‘4.9 Exosomes and Microvescicles isolation by differential ultracentrifugation’ (line 401-414). The RT-qPCR analysis has been performed using an internal control of the kit, as a reference. We introduced a new section ‘4.12 Analysis of miR-153 secretion in exosomes and microvesicles’ within the Material and Methods section, line 452-460 of the revised manuscript.

Reviewer 2 Report

The Authors indicate that miR-153 is secreted in PC cell culture fluids,  both  by  microvesicles  and  exosomes, what may influence the surrounding tissues. Interestingly, in two tumor lines of the prostate, the advantage of different secretion mechanisms (microvesicles or exosomes). the work is very interesting, well organized and very well written, and may undoubtedly influence the development of prostate cancer diagnostics. However, there are some ambiguities that I would like to be cleared up:

  • throughout the work, the culture density and the number of cells that are plated on the plates should be of the same form (10E3 or 10000 cells/well),
  • please provide ATCC numbers of cell lines, providing only the name of the cell line does not identify the line with Gleason score,
  • whether the BPH line is from the primary line?

Author Response

REVIEWER 2

Comments and Suggestions for Authors

The manuscript entitled:"Secreted miR-153 controls proliferation and invasion of higher Gleason Score prostate cancer" focused on the evaluation of predictive role of miR-153 in prostate cell cancer models is well written and requires minor revisions to be suitable for publication:

  1. In the introduction section, please, could the authors report if other miRNA have a potential predictive role in the managment of prostate cancer patients?

We thank the reviewer for the request. We added other miRNAs with a potential predictive role in PC in the ‘Introduction’ section, line 70-73.

  1. In the material and method section, please, could the authors better describe real sample population cohort enrolled in this study? In my opinion, this aspect is functional to qualify the clinical stage where this approach may be adopted.

We thank the reviewer for the request. The UALCAN analysis includes all the samples of the TCGA database, containing 497 prostate cancer (Gleason score 1-9) and 52 normal samples. The PC samples include adenomas, adenocarcinomas, ductal and lobular neoplasms, cystic, mucinous and serous neoplasms. TCGA database also contains the information about the Gleason score, as reported in the manuscript. Details have been added into 4. Materials and Methods section, ‘4.3 UALCAN analysis’, line 346-348.

  1. In the results section, please, could the authors show if they have confirmed exosomes extraction by performing focused experimental procedures?

We thank the reviewer for the request. Exosome purification was performed by differential ultracentrifugation as described in the Materials and Methods section. The 16000xg pellet (microvesicles) and the 100000xg pellet (exosomes), were tested by western blot for the presence of TSG101, a specific exosome marker protein. Only the 100000xg is positive for TSG101, indicating that exosomes are correctly isolated and not present in the microvesicle fraction. (See new Fig.9, panel A, line 247-248, and legend lines 250-253).

  1. In the discussion section, please, could the authors evaluate if an integrating miRNA pool may improve the clinical relevance of this approach?

We thank the reviewer for the request. We added a sentence on this theme in ‘3.Discussion’ section, line 292-296.

Reviewer 3 Report

The manuscript entitled:"Secreted miR-153 controls proliferation and invasion of higher Gleason Score prostate cancer" focused on the evaluation of predictive role of miR-153 in prostate cell cancer models is well written and requires minor revisions to be suitable for publication:

  • In the introduction section, please, could the authors report if other miRNA have a potential predictive role in the managment of prostate cancer patients?
  • In the material and method section, please, could the authors better describe real sample population cohort enrolled in this study? In my opinion, this aspect is functional to qualify the clinical stage where this approach may be adopted.
  • In the results section, please, could the authors show if they have confirmed exosomes extraction by performing focused experimental procedures?
  • In the discussion section, pelase, could the authors evaluate if an integrating miRNA pool may improve the clinical relevance of this approach?

Author Response

REVIEWER 3

The Authors indicate that miR-153 is secreted in PC cell culture fluids, both by microvesicles and exosomes, what may influence the surrounding tissues. Interestingly, in two tumor lines of the prostate, the advantage of different secretion mechanisms (microvesicles or exosomes). the work is very interesting, well organized and very well written, and may undoubtedly influence the development of prostate cancer diagnostics. However, there are some ambiguities that I would like to be cleared up:

  • throughout the work, the culture density and the number of cells that are plated on the plates should be of the same form (10E3 or 10000 cells/well),

We thank the reviewer for the observation. We carefully check the whole manuscript and corrected uniformly the cell density number (see ‘4.Materials and Methods’ section, lines 363, 376 and 397 of the revised manuscript).

  • please provide ATCC numbers of cell lines, providing only the name of the cell line does not identify the line with Gleason score,

We thank the reviewer for this observation. We added the ATCC reference numbers of each cell line, as requested (see ‘4.Materials and Methods’ section, lines 311-312 of the revised manuscript).

Moreover, we selected these two cell lines for the following reasons:

-in ATCC characterization report, PC3 cell line is described as a grade IV adenocarcinoma. Gleason Grading system and recent changes from International Society of Urological Pathology consensus conference on Gleason Grading of Prostatic Carcinoma suggested that grade IV corresponds to Gleason score 8 [Gordetsky and Epstein, 2016, DOI: 10.1186/s13000-016-0478-2]

-DU145 are aggressive, invasive adenocarcinoma from brain metastatic lesion. They are reported to be a grade II adenocarcinoma (ATCC website), which correspond to Gleason score 7 [Gordetsky and Epstein, 2016 PUBMED].

We inserted this information in section ‘4.Material and Methods’, line 313-318 of the revised manuscript.

  • whether the BPH line is from the primary line?

We thank the reviewer for the question. It is an immortalized cell line from human benign prostate epithelial cells hyperplasia (68 years old donor). We added this information in section ‘4. Material and Methods’, line 318-319 of the revised manuscript.

Round 2

Reviewer 1 Report

Although the Authors have improved the manuscript there are still important points to be addressed.

1) The Authors measured the miR-153 in the PC-3 and DU-145 medium (compared to the intracellular counterpart) (new Figure 8A), but the crucial point that remains to be demonstrated is that miR-153 level in the PC-3/DU-145 medium is higher than in BPH1 medium, otherwise it is unlikely that the miR-153 increase after treatment with PC-3/DU-145 conditioned medium could depend on the “uptake” of miR-153 from the medium. It should be noted that higher expression of a miRNA does not always mean increased release in the culture medium. 

2) miR-153 behaves as an “onco-miRNA” in PCa cells, in that its downregulation (through upregulation of KLF5) results in decreased proliferation, migration and invasion. In contrast, in BPH1 cells, miR-153 behaves as a “TS-miRNA” (again by regulating KLF5), in that its increase due to the treatment of PCa cells CM results in decreased proliferation. I think that this point should be clearly stated in the discussion and also more extensively discussed. Moreover, I believed that the Authors should provide a possible explanation of the biological significance of this datum.  

Minor point

For the luciferase assay, the CXCR4 target gene does not appear to be included in the pEZX-MT05 vector. What is the “HmiT111619-MT05” construct? Which vector did the Authors use for the assay? This point should be clarified. 

Author Response

Reviewer 1 – 27th May 2022

Comments and Suggestions for Authors

Although the Authors have improved the manuscript there are still important points to be addressed.

  • The Authors measured the miR-153 in the PC-3 and DU-145 medium (compared to the intracellular counterpart) (new Figure 8A), but the crucial point that remains to be demonstrated is that miR-153 level in the PC-3/DU-145 medium is higher than in BPH1 medium, otherwise it is unlikely that the miR-153 increase after treatment with PC-3/DU-145 conditioned medium could depend on the “uptake” of miR-153 from the medium. It should be noted that higher expression of a miRNA does not always mean increased release in the culture medium. 

We thank the reviewer for this observation. We performed the requested experiment, comparing the level of expression of miR-153 in conditional medium of BPH1 compared to those of PC3 and DU145 conditional media. The results are presented in the new figure 8, Line235-236 of the Legend and Results section, paragraph 2.5 miR-153 could be secreted, influencing surrounding tissue. Lines 216-221 ; they have been also discussed in Discussion section, lines 290-291.

  • miR-153 behaves as an “onco-miRNA” in PCa cells, in that its downregulation (through upregulation of KLF5) results in decreased proliferation, migration and invasion. In contrast, in BPH1 cells, miR-153 behaves as a “TS-miRNA” (again by regulating KLF5), in that its increase due to the treatment of PCa cells CM results in decreased proliferation. I think that this point should be clearly stated in the discussion and also more extensively discussed. Moreover, I believed that the Authors should provide a possible explanation of the biological significance of this datum.  

We thank the reviewer for the observation, we discuss this point in the Discussion section, line 292-302 of the Revised Manuscript.

Minor point

For the luciferase assay, the CXCR4 target gene does not appear to be included in the pEZX-MT05 vector. What is the “HmiT111619-MT05” construct? Which vector did the Authors use for the assay? This point should be clarified. 

We thank the reviewer for this observation: the HmiT111619-MT05” construct is the pEZX-MT05 vector (GeneCopeia) in which the 3’UTR of CXCR4 was cloned by EcoRI/XhoI sites (GeneCopeia). In order to generate the construct containing 3’UTR of KLF5, we cut out the fragment of the 3’UTR of CXCR4 and cloned in the 3’UTR of KLF5 by EcoRI/XhoI cloning sites. We clarify this point in Material and Methods section, par. 4.4 Luciferase reporter assay, lines 367-372 of the Revised Manuscript.

Round 3

Reviewer 1 Report

The Authors' responses to my comments were quite sufficient